# New Diterpenes with a Fused 6-5-6-6 Ring System Isolated from the Marine Sponge-Derived Fungus *Trichoderma harzianum*

**DOI:** 10.3390/md17080480

**Published:** 2019-08-19

**Authors:** Takeshi Yamada, Ayano Fujii, Takashi Kikuchi

**Affiliations:** Department of Medicinal Molecular Chemistry, Osaka University of Pharmaceutical Sciences, 4-20-1, Nasahara, Takatsuki, Osaka 569-1094, Japan

**Keywords:** diterpenes, trichodermanins, *Trichoderma harzianum*, *Halichondria okadai*, fused 6-5-6-6 ring system, cytotoxicity

## Abstract

New diterpenes, namely, trichodermanins F–H, with a fused 6-5-6-6 ring system were isolated from the fungus *Trichoderma harzianum* OUPS-111D-4 separated from a marine sponge *Halichondria okadai*. These chemical structures were elucidated by 1D and 2D NMR as well as high-resolution fast atom bombardment mass spectrometry (HRFABMS) spectral analyses. We established their absolute stereostructures by application of the modified Mosher’s method or circular dichroism (CD) spectroscopy. In addition, their cytotoxicities were assessed using several cancer cell lines, with **1** and **2** exhibiting modest activities.

## 1. Introduction

A number of secondary metabolites derived from marine microorganisms have diverse structures and exhibit unexpected biological activities [1,2,3,4]. This motivated us to target marine microorganisms as a source of seed compounds for antitumor chemotherapy agents, and we have reported many cytotoxic metabolites to date [5,6,7,8]. As part of this ongoing study, we examined the metabolites of the fungus *Trichoderma harzianum* separated from a piece of the marine sponge *Halichondria okadai*. We have already reported the isolation, structural determination, and cytotoxicity of the decalin derivatives tandyukisins A–F derived from this fungus [9,10,11] as well as those of trichodermanins C, D, and E (**1**), classified as diterpenes with a rare fused 6-5-6-6 ring system [12]. In the continuing search for cytotoxic metabolites from this fungal strain, we isolated three new compounds, namely, trichodermanins F–H (**2**–**4**). Except for our study, there are only two other reports of natural products consisting of this ring system, i.e., trichodermanin A [13] and wickerols A and B [14]. We have previously reported the first determination of the absolute configurations of trichodermanins C and D by application of the modified Mosher’s method [15]. In the present study, we established the absolute configuration of trichodermanin E (**1**), which was confirmed by circular dichroism (CD) spectroscopy. In addition, we also examined the absolute stereostructures of the newly isolated trichodermanins F–H (**2**–**4**). We herein describe the evaluation of the cytotoxic activities of **1**–**4**, although they exhibited zero or weak activities.

## 2. Results and Discussion

*T. harzianum*, a fungus from *H. okadai*, was cultured at 27 °C for six weeks in a medium (70 L) containing 1% glucose, 1% malt extract, and 0.05% peptone in artificial seawater adjusted to pH 7.5. After incubation, the EtOAc extract of the culture filtrate was purified via cytotoxic assay-directed fractionation using a stepwise combination of silica gel column and Sephadex LH-20 chromatography, followed by reverse-phase HPLC, affording trichodermanins E (**1**) (0.93 mg, 0.011%), F (**2**) (3.7 mg, 0.043%), G (**3**) (4.6 mg, 0.053%), and H (**4**) (1.3 mg, 0.015%) 

The molecular formula of trichodermanin E (**1**) was established as C_20_H_34_O_3_ based on the molecular ion *m*/*z* 345.2397 [M + Na]^+^ in high-resolution fast atom bombardment mass spectrometry (HRFABMS) (Appendix A). Absorption in the IR spectrum at 3383 cm^−1^ indicated the presence of hydroxy groups. The relative stereostructure of **1** has previously been reported using distortionless enhancement by polarization transfer (DEPT), heteronuclear multiple quantum coherence spectroscopy (HMQC), and NOESY experiments with the ^1^H and ^13^C NMR spectra of **1** (Figure 1, and Appendix A, Table 1, and Appendix A) [12]. However, the assignment of its absolute configuration was not performed. In the present study, the absolute stereostructure of **1** was elucidated by acylation with *p*-bromobenzoyl chloride to apply the dibenzoate rule [16]. In the CD spectrum of the dibenzoate **1a**, the negative Cotton effect (Δ*ε*_258_–15) demonstrated that the screw sense of two benzoate chromophores at C-1 and C-2 was counterclockwise. Therefore, the absolute configurations for C-1 and C-2 were revealed as *R* and *R*, respectively (Figure 2) [16].

Trichodermanin F (**2**) was assigned the formula C_20_H_34_O_2_ by HRFABMS *m*/*z* 289.2534 [M − Na]^+^ (calcd for C_20_H_34_O_3_Na: 289.2532), which contained one less oxygen atom than that of **1** (Appendix A). Consideration of the ^1^H and ^13^C NMR spectra of **2** (Table 1 and Appendix A), including 2D NMR spectra, suggested that compound **2** had one secondary methyl (C-17); four tertiary methyls (C-16, C-18, C-19, and C-20); six sp^3^-hybridized methylenes (C-1, C-7, C-9, C-10, C-13, and C-14); five sp^3^ methines (C-2, C-3, C-6, C-11, and C-12), including one oxygen-bearing sp^3^ methine (C-2); and four quaternary sp^3^ carbons (C-4, C-5, C-8, and C-15), including one oxygen-bearing quaternary sp^3^ carbon (C-15). ^1^H–^1^H correlation spectroscopy (COSY) revealed three partial structures (Figure 3 and Appendix A). In the HMBC spectrum (Figure 3 and Appendix A), the correlation from 15-methyl to C-11, C14, and C-15; from 8-methyl to C-7, C-8, C-9, and C-12; from germinal dimethyl (H-18 and H-19) to C-4, C-5, and C-6; from H-12 to C-3, C-4, and C-5; and from H-13 to C-12 showed three cyclohexane rings and a cyclopentane comprising a fused 6-5-6-6 ring system to give the planar structure of **2**, as shown in Figure 3.

For the stereochemistry of **2**, the relative configuration and conformation were investigated by NOESY experiments (Appendix A and Figure 4 and Appendix A). In the NOESY analysis of **2**, correlations from H-2α to H-19; from H-3 to H-11, H-14β, and H-20; and from H-1β to H-20 indicated that cyclohexane ring A existed in the boat conformation with 3-CH_3_ in the α-orientation. In cyclohexane ring B, the above correlations and the NOESY cross peaks (H-7α/H-12 and H-18) demonstrated that ring B existed in the chair conformation and that 5-CH_3_ (C-18), H-7α, and H-12 were oriented in coaxial arrangements. The NOESY correlations between H-3, H-11, and H-14β indicated that cyclohexane ring C also existed in the chair conformation. In addition, the correlations (H-18/H-13α, H-11/H-20, H-14α/H-16, and H-12/H-16) revealed that H-11, H-13α, H-14β, and 15-CH_3_ (C-16) were oriented in coaxial arrangements and that the ring juncture between cyclohexane rings B and C was *trans* (Figure 4 and Appendix A). 

A difference in the structural features of **2** and **1** was the absence of a secondary hydroxyl group at C-1. Therefore, we applied the modified Mosher’s method [15] to determine their absolute stereostructures. The ^1^H chemical shift differences between the (*S*)- and (*R*)-MTPA esters **2a** and **2b** showed positive on the right side and negative on the left side of the MTPA plane; therefore, we revealed an *S* configuration at C-2 (Figure 5 and Appendix A).

Trichodermanin G (**3**) was assigned the formula C_20_H_34_O_3_ by HRFABMS *m*/*z* 345.2395 [M + Na]^+^ (calcd for C_20_H_34_O_3_Na: 345.2396), which was the same as that of **1** (Appendix A). The NMR spectral features (Table 1 and Appendix A) resembled those of **1**, except for the proton signals for H-2 (*δ*_H_ 1.45, m and *δ*_H_ 2.66, ddd), H-3 (*δ*_H_ 2.10, m), and H-10 (*δ*_H_ 4.38, ddd) and the carbon signals of C-1 (*δ*_C_ 72.6), C-2 (*δ*_C_ 41.8), C-3 (*δ*_C_ 26.7), C-9 (*δ*_C_ 52.2), C-10 (*δ*_C_ 72.9), and C-11 (*δ*_C_ 55.0) in **3**. The ^1^H–^1^H COSY correlations (H-1/H-6, H-1/H-2, and H-2/H-3), and HMBC correlation from H-17 to C-2 indicated that the hydroxy methine at C-2 in **1** were replaced by a methylene in **3** (Table 1 and Appendix A). In addition, the ^1^H–^1^H COSY correlations (H-9/H-10, H-10/H-11, and H-11/H-12) and HMBC correlations (H-20/C-9, H-12/C-10, and H-16/C-11) demonstrated that the methylene at C-10 in **1** were replaced by a hydroxy methine in **3** (Table 1 and Appendix A). The HMBC correlations for the 6-5-6-6 skeleton were the same as those of **1** and **3** (Appendix A). The above evidence confirmed the planar structure of **3**. The study for the stereochemistry of **3** is described later (together with that of **4**).

Trichodermanin H (**4**) had the same molecular formula C_20_H_34_O_3_ by HRFABMS *m*/*z* 345.2415 [M + Na]^+^ (calcd for C_20_H_34_O_3_Na: 345.2396) as that of **1** and **3** (Appendix A). Comparing the NMR spectra of **4** with those of **3** (Table 1, Appendix A), we proposed that the differences in the NMR chemical shifts at H-3 (*δ*_H_ 1.88), H-10 (*δ*_H_ 1.59, 1.80), H-17 (*δ*_H_ 3.60, 3.95), C-2 (*δ*_C_ 36.9), C-3 (*δ*_C_ 34.0), C-9 (*δ*_C_ 43.5), C-10 (*δ*_C_ 21.6), C-11 (*δ*_C_ 44.1), and C-17 (*δ*_C_ 68.1) for **4** from those for **3** were caused by a change in the linkage position of the two hydroxy groups, i.e., **4** was a positional isomer of **3**. The NMR spectral features (Table 1, Appendix A) resembled those of **1**, except for the proton signals for H-2 (*δ*_H_ 2.74, ddd and *δ*_H_ 1.64, ddd) and H-17 (*δ*_H_ 3.60, dd and 3.95, dd) and the carbon signals of C-1 (*δ*_C_ 72.1) and C-17 (*δ*_C_ 68.1) in **4**. These data and the ^1^H–^1^H COSY correlation between H-3 and H-17 and H-9 and H-10 indicated the planar structure of **2** (Appendix A). This structure was supported by the HMBC correlations observed being similar to those of **1**, **2**, and **3** (Appendix A). The most remarkable structural feature of **4** was that 3-CH_3_ (C-17) in the previous trichodermanins was replaced with a hydroxy methyl group. 

The relative stereochemical configurations and conformations of **3** and **4** were investigated by NOESY experiments (Appendix A, Figure 6, and Appendix A). The key NOESY correlations for ring A (H-2α/H-19, H-1/H20, and H-3/H-20), ring B (H-7α/H-12, H-7α/H-18, and H-12/H-18), and ring C (H-3/H-11, H-11/H-14β, and H-13α/H-18) showed that the relative configurations and conformations in the 6-5-6-6 ring system of **3** and **4** were identical with those of the above metabolites (Appendix A, Figure 6, and Appendix A). The absolute configurations of **3** and **4** were not positively elucidated because of the small amount of material available. However, it is unreasonable to assume that only **3** and **4** derived from the same fungus are enantiomers of other trichodermanins. Therefore, we suggest the absolute stereostructures of **3** and **4** as shown in Figure 1 based on assumed biosynthetic relatedness. These stereostructural hypotheses will be confirmed using the modified Mosher’s method [15], the quantum chemical equivalent circulating density (ECD) calculation method [17], or synthetic research in the future.

As a primary screen for antitumor activity, the cancer cell growth inhibitory properties of trichodermanins E–H (**1**–**4**) were examined using murine P388 leukemia, human HL-60 leukemia, and murine L1210 leukemia cell lines. The results are shown in Table 2. Both **1** and **2** exhibited modest activities against the cell lines, although they were inferior to that of 5-fluorouracil (Table 2). However, **3** and **4** did not inhibit cell growth at all (Table 2). In the study of the structure–activity relationship inferred from the activities of **1**–**4** and previous trichodermanins E and D, the binding positions (C-1, C-2, C-6, C-10, and C-17) of the hydroxy groups did not contribute to the enhancement of activity. The number had some influence, i.e., the activities of diol derivatives as **2** and trichodermanins C were more potent than those of the triol derivatives as **1**, **3**, **4**, and trichodermanin D. In particular, trichodermanins C with a carbonyl group at C-1 showed strong activity comparable to 5-fluorouracil. We speculate that the significant cell growth inhibition of trichodermanins C was caused by the change of conformation of A ring due to the carbonyl group at C-1. A comparison of cytotoxicity with normal cell and an elucidation of the detailed mechanism of activity will continue to be examined after supply of these fungal metabolites.

## 3. Materials and Methods 

### 3.1. General Experimental Procedures

These are the same procedures as those in recent reports [7,8,9,10,11]. NMR spectra were recorded on an Agilent-NMR-vnmrs (Agilent Technologies, Santa Clara, CA, USA) 600 with tetramethylsilane (TMS) as an internal reference. FABMS was recorded using a JEOL JMS-7000 mass spectrometer (JEOL, Tokyo, Japan). IR spectra were recorded on a JASCO FT/IR-680 Plus (Tokyo, Japan). Optical rotations were measured using a JASCO DIP-1000 digital polarimeter (Tokyo, Japan). Silica gel 60 (230–400 mesh, Nacalai Tesque, Inc., Kyoto, Japan) was used for column chromatography with medium pressure. Octadecyl silica (ODS) HPLC was run on a JASCO PU-1586 (Tokyo, Japan) equipped with a differential refractometer RI-1531 (Tokyo, Japan) and Cosmosil packed column 5C18-MSII (25 cm × 20 mm i.d., Nacalai Tesque, Inc., Kyoto, Japan). Analytical thin-layer chromatography (TLC) was performed on precoated Merck aluminum sheets (DC-Alufolien Kieselgel 60 F254, 0.2 mm, Merck, Darmstadt, Germany) with the solvent system CH_2_Cl_2_/MeOH (19:1) (Nacalai Tesque, Inc., Kyoto, Japan), and compounds were viewed under a UV lamp (AS ONE Co., Ltd., Osaka, Japan) and sprayed with 10% H_2_SO_4_ (Nacalai Tesque, Inc., Kyoto, Japan), followed by heating.

### 3.2. Fungal Material

This study is a follow-up report for this fungal strain. As shown in the previous reports [9], the fungus *Trichoderma harzianum* was isolated from a piece of inner tissue of the marine sponge *Halichondria okadai* collected at Osaka Bay, Japan, in October 2008. The fungal strain was identified by Techno Suruga Laboratory Co., Ltd. The sponge was wiped with EtOH and a cutting applied to the surface of nutrient agar layered in a Petri dish. Serial transfers of one of the resulting colonies provided a pure strain of *T. harzianum*.

### 3.3. Culturing and Isolation of Metabolites

The EtOAc extract (8.6 g) of the culture filtrate, which was obtained as described in previous reports [9,10,11,12], was chromatographed on a silica gel column with a CHCl_3_/MeOH gradient as the eluent to afford Fr. 1 (5% MeOH in CHCl_3_ eluate, 65.6 mg) and Fr. 2 (5% MeOH in CHCl_3_ eluate, 119.7 mg). Fr. 1 was purified by ODS HPLC using MeOH/H_2_O (80:20) as the eluent to afford Fr. 3 (11.3 mg) and **2** (3.7 mg, *t_R_* 60 min). Fr. 3 was purified by HPLC using MeCN/H_2_O (40:60) as the eluent to afford **1** (0.93 mg, *t_R_* 27.5 min). Fr. 2 was purified by ODS HPLC using MeOH/H_2_O (80:20) as the eluent to afford Fr. 4 (9.9 mg). Fr. 4 was purified by HPLC using MeCN/H_2_O (30:70) as the eluent to afford **3** (0.8 mg, *t_R_* 58 min) and **4** (1.3 mg, *t_R_* 60 min)

Trichodermanins E (**1**): pale yellow oil; [α]${}_{D}^{22}$ + 188.0 (*c* 0.09, MeOH); IR (neat) *ν*_max_/cm^−1^: 3383. HRFABMS *m*/*z* 345.2397 [M + Na]^+^ (calcd for C_20_H_34_O_3_Na: 345.2396); NMR data, see Table 1 and Appendix A and the previous report [12].

Trichodermanins F (**2**): pale yellow oil; [α]${}_{D}^{22}$ + 22.4 (*c* 0.24, MeOH); IR (neat) *ν*_max_/cm^−1^: 3398. HRFABMS *m*/*z* 289.2534 [M − Na]^+^ (calcd for C_20_H_34_O_3_Na: 289.2532). NMR data, see Table 1 and Appendix A.

Trichodermanins G (**3**): pale yellow oil; [α]${}_{D}^{22}$ + 8.0 (*c* 0.08, MeOH); IR (neat) *ν*_max_/cm^−1^: 3387. HRFABMS *m*/*z* 345.2395 [M + Na]^+^ (calcd for C_20_H_34_O_3_Na: 345.2396). NMR data, see Table 1 and Appendix A.

Trichodermanin H (**4**): pale yellow oil; [α]${}_{D}^{22}$ + 52.0 (*c* 0.04, MeOH); IR (neat) *ν*_max_/cm^−1^: 3359. FABMS *m*/*z* (rel. int.): 345 ([M + Na]^+^, 37.4%) 305 (43.8%), 287 (54.5%), 115 (100%). HRFABMS *m*/*z* 345.2415 [M + Na]^+^ (calcd for C_20_H_34_O_3_Na: 345.2396). NMR data, see Table 1 and Appendix A.

### 3.4. Formation of Dibenzoate of 1

To a solution of **1** (1.8 mg, 5.6 μmol) in abs. pyridine (0.3 mL), *p*-bromobenzoyl chloride (3.0 mg, 13.6 μmol) was added, and the reaction mixture was stirred at room temperature for 15 h. Water (1.0 mL) was added to the reaction mixture and extracted using CH_2_Cl_2_. The organic layer was evaporated under reduced pressure, and the residue was purified by HPLC using 100% MeOH as the eluent to afford dibenzoate **1a** (2.2 mg, 58.8%) as a pale yellow oil.

Dibenzoate **1a**: pale yellow oil; HRFABMS *m*/*z* 709.1138 [M + Na]^+^ (calcd for C_34_H_40_O_5_^79^Br_2_Na: 709.1139). ^1^H NMR data (TMS, CDCl_3_) *δ* ppm: 1.04 (3H, s), 1.13, 1.21 (3H, d), 1.22 (3H, s), 1.23 (3H, s), 1.24 (3H, s), 1.26 (3H, s), 1.28, 1.47, 1.56, 1.58, 1.68, 1.82, 1.85, 1.88, 1.92, 2.07, 2.25, 5.50 (1H, d), 5.85 (1H, dd), 7.56 (Ar. H), 7.81 (Ar. H), 7.88 (Ar. H).

### 3.5. Formation of the (S)- and (R)-MTPA Esters of 2

To a solution of **2** (1.0 mg, 3.3 μmol) in abs. pyridine (0.3 mL), (*R*)-(–)-MTPA chloride (3.0 mg, 15.7 μmol) was added, and the reaction mixture was stirred at room temperature for 2 h. The reaction mixture was evaporated under reduced pressure, and the residue was purified by HPLC using MeOH–H_2_O (90:10) as the eluent to afford (*S*)-MTPA ester **2a** (0.8 mg, 47.2%).

**2** (1.2 mg, 3.9 μmol) and (*S*)-(+)-MTPA chloride (3.0 mg, 15.7 μmol) were treated with the same procedure to afford (*R*)-MTPA ester **2b** (1.5 mg, 73.7%).

MTPA ester **2a**: pale yellow oil; HRFABMS *m*/*z* 545.2860 [M + Na]^+^ (calcd for C_30_H_41_F_3_O_4_Na: 545.2855) (Appendix A). ^1^H NMR data and spectrum including 2D NMR spectra are listed in Appendix A.

MTPA ester **2b**: pale yellow oil; HRFABMS *m*/*z* 545.2849 [M + Na]^+^ (calcd for C_30_H_41_F_3_O_4_Na: 545.2855) (Appendix A). ^1^H NMR data and spectrum including 2D NMR spectra are listed in Appendix A.

### 3.6. Assay for Cytotoxicity

Cytotoxic activities of **1**–**4** were examined by the same procedure to date [5,6,7,8,9,10,11,12,18], i.e., the 3-(4,5-dimethyl-2-thiazolyl)-2,5-diphenyl- 2H-tetrazolium bromide (MTT) method. P388, HL-60, and L1210 cells were cultured in RPMI 1640 Medium (10% fetal calf serum) at 37 °C in 5% CO_2_. The test materials were dissolved in dimethyl sulfoxide (DMSO) to give a concentration of 10 mM, and the solution was diluted with the Essential Medium to yield concentrations of 200, 20, and 2 μM, respectively. Each solution was combined with each cell suspension (1 × 10^5^ cells/mL) in the medium, respectively. After incubating at 37 °C for 72 h in 5% CO_2_, grown cells were labeled with 5 mg/mL MTT in phosphate-buffered saline (PBS), and the absorbance of formazan dissolved in 20% sodium dodecyl sulfate (SDS) in 0.1 N HCl was measured at 540 nm with a microplate reader (MTP-310, CORONA electric). Each absorbance value was expressed as percentage relative to that of the control cell suspension that was prepared without the test substance using the same procedure as that described above. All assays were performed three times, semilogarithmic plots were constructed from the averaged data, and the effective dose of the substance required to inhibit cell growth by 50% (IC_50_) was determined. 

### 3.7. The Origin of the Cell Lines

P388 cell line was obtained from Dr. Numata. HL-60 cell line was obtained from Dr. Kawai. L1210 cell line was from Dr. Endo.

## 4. Conclusions

In this study, three new terpenes with a rare fused 6-5-6-6 ring system, namely, trichodermanins E–H (**1**–4), were isolated from the fungus *T. harzianum* separated from the marine sponge *H**. okadai*. Spectral analyses and chemical transformations were utilized to elucidate the absolute stereostructures of **1** and **2**. The relative stereostructures of **3** and **4** were assumed to be those of the same class of metabolites. Therefore, we deduced that their absolute configurations were also the same as those of other trichodermanins. In cytotoxic assays performed using three cancer cell lines, **1** and **2** exhibited moderate activity against the cell lines, although they were inferior to that of 5-fluorouracil.

## Figures and Tables

**Figure 1 marinedrugs-17-00480-f001:**
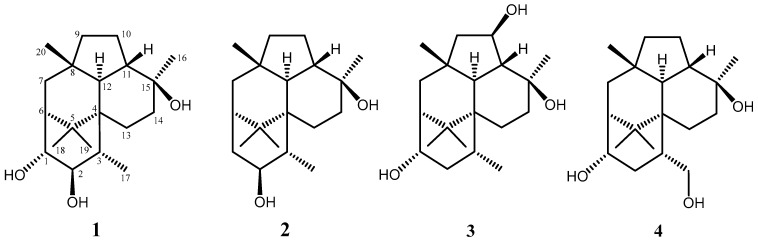
Structures of trichodermanins E–H (**1**–**4**).

**Figure 2 marinedrugs-17-00480-f002:**
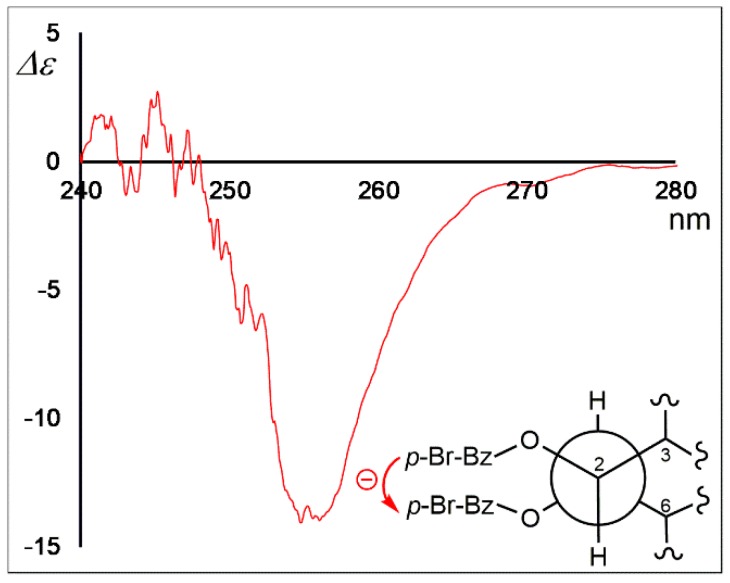
Circular dichroism (CD) spectrum of the dibenzoate of trichodermanin E (**1a**).

**Figure 3 marinedrugs-17-00480-f003:**
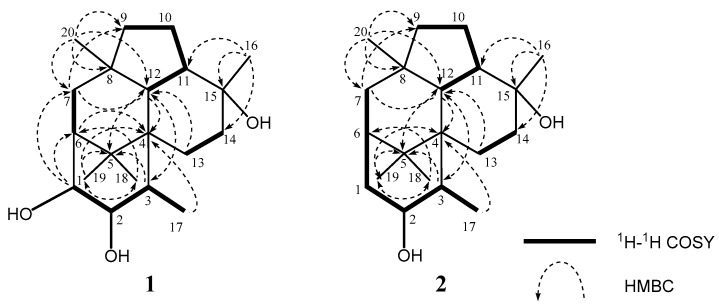
Key ^1^H–^1^H COSY and HMBC correlations for (**1**,**2**).

**Figure 4 marinedrugs-17-00480-f004:**
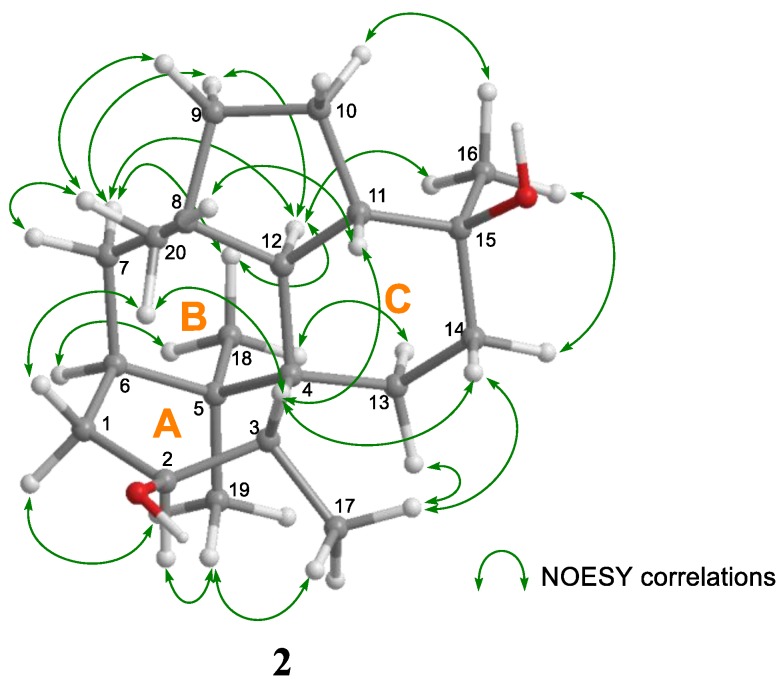
Key NOESY correlations for **2**.

**Figure 5 marinedrugs-17-00480-f005:**
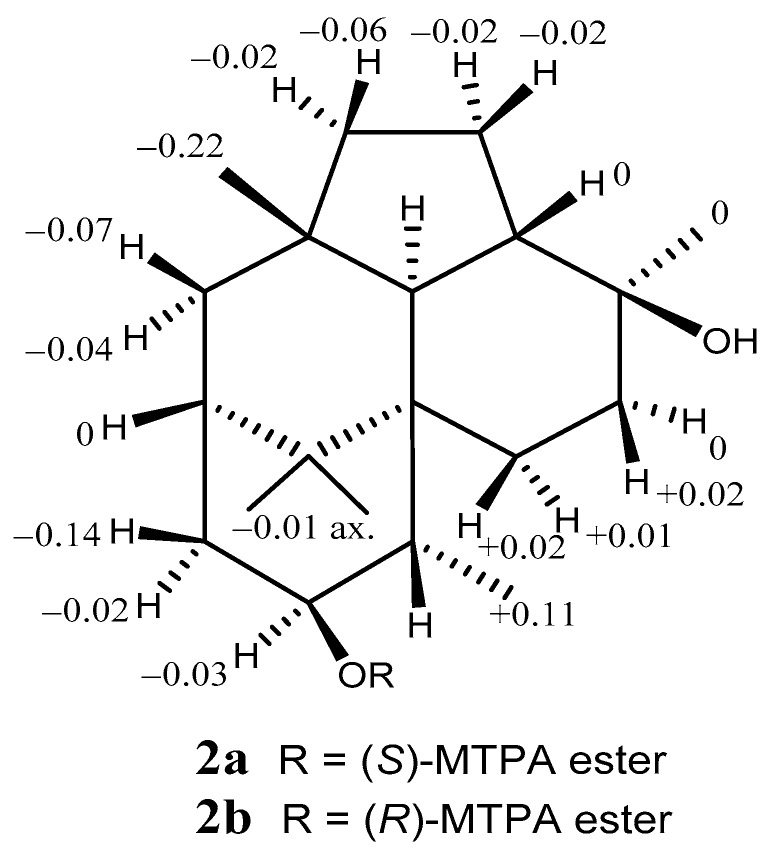
^1^H chemical-shift differences between the (*S*)- and (*R*)-MTPA esters **2a** and **2b**.

**Figure 6 marinedrugs-17-00480-f006:**
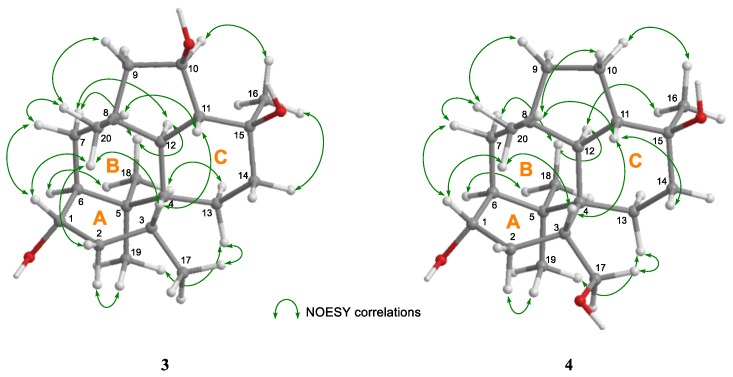
Key NOESY correlations for **3** and **4.**

**Table 1 marinedrugs-17-00480-t001:** NMR spectral data for **1–4** in CDCl_3._

Position	1				2				3				4			
	*δ* _H_ *^a^*		*δ* _C_		*δ* _H_ *^a^*		*δ* _C_		*δ* _H_ *^a^*		*δ* _C_		*δ* _H_ *^a^*		*δ* _C_	
1α					2.54	ddd	36.5	(t)								
1β	4.11	d	80.4	(d)	1.68	m			4.20	dd	72.6	(d)	4.24	dd	72.1	(d)
2α	3.88	dd	83.7	(d)	4.30	ddd	74.2	(d)	2.66	ddd	41.8	(t)	2.74	ddd	36.9	(t)
2β									1.45	m			1.64	ddd		
3	1.88	qd	36.6	(d)	1.98	qd	37.9	(d)	2.10	m	26.7	(d)	1.88	dddd	34.0	(d)
4			41.2	(s)			40.8	(s)			38.9	(s)			38.9	(s)
5			39.4	(s)			39.0	(s)			39.0	(s)			38.5	(s)
6	1.50	dd	53.2	(d)	1.62	m	41.8	(d)	1.41	dd	52.2	(d)	1.45	dd	52.2	(d)
7α	1.78	dd	40.9	(t)	1.71	dd	42.5	(t)	1.69	dd	41.3	(t)	1.74	dd	40.9	(t)
7β	1.70	dd			1.56	dd			1.65	dd			1.68	dd		
8			39.6	(s)			39.7	(s)			38.8	(s)			39.2	(s)
9α	1.03	m	43.5	(t)	1.02	m	43.9	(t)	1.48	m	52.2	(t)	1.03	m	43.5	(t)
9β	1.43	m			1.43	m							1.43	m		
10α	1.59	m	21.6	(t)	1.60	m	21.6	(t)	4.38	ddd	72.9	(d)	1.59	m	21.5	(t)
10β	1.80	m			1.81	m							1.80	m		
11	1.81	dd	44.2	(d)	1.85	m	44.4	(d)	1.85	dd	55.0	(d)	1.88	ddd	44.1	(d)
12	1.32	d	51.8	(d)	1.30	d	51.9	(d)	1.28	d	50.6	(d)	1.28	d	52.0	(d)
13α	1.23	ddd	26.3	(t)	1.18	ddd	26.4	(t)	1.19	ddd	25.9	(t)	1.28	m	25.5	(t)
13β	1.72	ddd			1.68	m			1.74	ddd			1.52	m		
14α	1.64	ddd	41.1	(t)	1.62	m	41.2	(t)	1.60	ddd	41.0	(t)	1.67	m	40.87	(t)
14β	1.46	ddd			1.46	ddd			1.50	ddd			1.50	m		
15			73.6	(s)			73.8	(s)			73.1	(s)			73.6	(s)
16	1.18	s	20.5	(q)	1.18	s	20.4	(q)	1.22	s	21.5	(q)	1.18	s	20.5	(q)
17	1.23	d	20.0	(q)	1.19	d	20.4	(q)	1.09	d	22.4	(q)	3.60	dd	68.1	(t)
													3.95	dd		
18ax	0.99	s	25.7	(q)	0.95	s	25.3	(q)	0.97	s	25.8	(q)	0.99	s	25.7	(q)
19eq	1.04	s	25.2	(q)	0.96	s	25.4	(q)	1.14	s	24.9	(q)	1.13	s	25.5	(q)
20	0.98	s	19.8	(q)	1.13	s	20.3	(q)	1.17	s	21.4	(q)	0.92	s	19.5	(q)

*^a^*^1^H chemical shift values (*δ* ppm from SiMe_4_) followed by multiplicity.

**Table 2 marinedrugs-17-00480-t002:** Cytotoxicity of metabolites **1**–**3** against cancer cell lines.

Compounds	Cell line P388	Cell line HL-60	Cell line L1210
	IC_50_ (μM) *^a^*	IC_50_ (μM) *^a^*	IC_50_ (μM) *^a^*
**1**	52.1 ± 1.3	59.8 ± 2.2	125.2 ± 4.3
**2**	58.9 ± 1.2	42.9 ± 3.0	41.5 ± 2.5
**3**	>300	>300	>300
**4**	>300	85.3 ± 2.1	73.2 ± 2.2
DMSO (control)	>300	>300	>300
5-fluorouracil *^b^*	3.9 ± 0.6	3.7 ± 0.1	4.2 ± 0.4

*^a^* DMSO was used as vehicle. *^b^* Positive control.

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
