# Peer review of "New Diterpenes with a Fused 6-5-6-6 Ring System Isolated from the Marine Sponge-Derived Fungus Trichoderma harzianum"

_marinedrugs, 2019, doi:10.3390/md17080480_

Round 1

Reviewer 1 Report

I appreciate the effort made by the Authors aimed at improving the quality of the revised version of the manuscript, according to my suggestions and criticisms. However, I really do not think that this manuscript reaches the required standards of originality, novelty and scientific soundness of the content to deserve publication, at least not in Marine Drugs. 

Summarizing, only the absolute configuration of Trichodermanins E (which was already tentatively assigned in the previous work Marine Drugs, 2017, 15, 169), while the absolute configuration of compounds 3 and 4 were not elucidated and only tentatively assigned. 

Basically, all the compounds did non exhibit significant cytotoxic effetcs in leukemic cell lines while cytotoxicity in normal cell lines is still unknown. Possible mechanism of action and biological target are undisclosed.  

Why the Authors think that their work deserves publication in Marine Drugs if new compunds are inactive, with unknown mechanism of action and with a putative absolute configuration?

I would suggest the submission of this work in Marine Drugs after resupply of these metabolites in suitable amount for structural elucidation and full biological appraisals or to consider the submission in a different journal.

I am really sorry about this.

Author Response

Reviewer #1

I appreciate the effort made by the Authors aimed at improving the quality of the revised version of the manuscript, according to my suggestions and criticisms. However, I really do not think that this manuscript reaches the required standards of originality, novelty and scientific soundness of the content to deserve publication, at least not in Marine Drugs. 

Summarizing, only the absolute configuration of Trichodermanins E (which was already tentatively assigned in the previous work Marine Drugs, 2017, 15, 169), while the absolute configuration of compounds 3 and 4 were not elucidated and only tentatively assigned. 

Basically, all the compounds did non exhibit significant cytotoxic effetcs in leukemic cell lines while cytotoxicity in normal cell lines is still unknown. Possible mechanism of action and biological target are undisclosed.  

Why the Authors think that their work deserves publication in Marine Drugs if new compunds are inactive, with unknown mechanism of action and with a putative absolute configuration?

I would suggest the submission of this work in Marine Drugs after resupply of these metabolites in suitable amount for structural elucidation and full biological appraisals or to consider the submission in a different journal.

Response: Thank you for your valuable opinions. We have challenged this journal with unique structures even though it has not exhibited significant bioactive. We will continue to do our best to respond to your comments. At this time, it is impossible to respond to your comments in terms of both sample quantity and laboratory technology, therefore; our second revision is our limit that we can. I would like to leave the results of this publish to the opinions of all reviewers.

Reviewer 2 Report

 Authors addressed all the points which we have questioned. I suggest to publish this manuscript in Marine drugs.

Author Response

Reviewer #2

Authors addressed all the points which we have questioned. I suggest to publish this manuscript in Marine drugs.

Response: Thank you for your evaluation.

Reviewer 3 Report

Overall, the manuscript reports interesting findings about new members of a previously reported family of diterpenes. 

Title - Suggest re-wording to:  New diterpenes with a fused 6-5-6-6 ring system isolated from the marine sponge-derived fungus Trichoderma Harzianum  

Abstract - add brief summary of cancer cell cytotox data.

 line 26  decaline to be replaced with decalin

line 38 replace microoroganism with fungus

line 40 identify the bioassay used to direct the purification

line 48 re-write to:  THe previously reported relative stereostructure... (ie. delete "the")

line 53 add ref for the dibenzoate rule 

line 61 (and for all compounds characterized) add observed and calculated exact masses and list  Δ ppm

line 63 re-word to:  spectra suggested that compound 2 has: one secondary methyl ... 

line 87-88 re-word to:  features of 2 and 1 was the absence of a secondary hydroxyl at C-1.  We applied the mosher's method...

line 90  end of paragraph needs a detailed discussion of how the stereochem at C-15 was deduced. 

line 93  the paragraph starting on line 93 needs to be rewritten.  Keep discussion of compounds 2 and 3 separate.  The description of 3 is not sufficiently detailed. 

line 106 remove "linkage"

line 113-114  last sentence of this paragraph does not make sense to me. all new compounds have the 6-5-6-6 ring system so how can compound 4 be the first?

line 122 re-word  Therefore, we deduced the absolute ...   The language should be softened - e.g. we suggest that the absolute stereochemistry of ....   based on assume biosynthetic relatedness. 

line 131  replace "moderate" with "modest"

line 138  replace "guessed" witih "speculate"

line 160  last word "The" should not be capitalized.

line 162  remove one of the "collected"

line 163  replace "its snip" with "a cutting"

lines 221 & 222 what is meant by "(death)"  Suggest these be deleted.

line 224  replace "four" with "three" as only 3 new compounds are reported - as stated in the abstract

line 227 replace "were identical" with "assumed to be"

Author Response

Reviewer #3

Overall, the manuscript reports interesting findings about new members of a previously reported family of diterpenes. 

Title - Suggest re-wording to:  New diterpenes with a fused 6-5-6-6 ring system isolated from the marine sponge-derived fungus Trichoderma Harzianum  

Response: As your pointed out, we re-wrote title.

Abstract - add brief summary of cancer cell cytotox data.

Response: We inserted brief summary of activity shown in red in Abstract

 line 26  decaline to be replaced with decalin

Response: Thank you for your pointing out, we corrected decaline to decalin.

line 38 replace microoroganism with fungus

Response: As be pointed out, we replaced microoroganism with fungus.

line 40 identify the bioassay used to direct the purification

Response: We corrected `bioassay’ to `cytotoxic assay’.

line 48 re-write to:  THe previously reported relative stereostructure... (ie. delete "the")

Response: Thank you for your pointing out, we deleted “the” before “relative stereostructure”.

Line 53 add ref for the dibenzoate rule 

Response: ref. no.16 was added in line 53. This is the same ref. as that in line 56.

line 61 (and for all compounds characterized) add observed and calculated exact masses and list  Δ ppm

Response: We added observed and calculated exact masses of all compounds.

line 63 re-word to:  spectra suggested that compound 2 has: one secondary methyl ... 

Response: As be pointed out, we re-wrote line 63. (ie, we deleted “the functional groups as below; i.e., this”.

line 87-88 re-word to:  features of 2 and 1 was the absence of a secondary hydroxyl at C-1.  We applied the mosher's method...

Response: Thank you for your pointing out, we re-wrote to “A difference in the structural features of 2 and 1 was the absence of a secondary hydroxyl group at C-1”.

line 90 end of paragraph needs a detailed discussion of how the stereochem at C-15 was deduced. 

Response: We inserted a sentence describing the distribution of differences.

line 93 the paragraph starting on line 93 needs to be rewritten.  Keep discussion of compounds 2 and 3 separate.  The description of 3 is not sufficiently detailed. 

Response: We rewrote the paragraph pointed out as shown in red. (ie, 2D correlation date were added). And I am sorry that We mistake compound 3 to compound 2.

line 106 remove "linkage"

Response: We removed "linkage".

line 113-114 last sentence of this paragraph does not make sense to me. all new compounds have the 6-5-6-6 ring system so how can compound 4 be the first?

Response: As be pointed out, this sentence does not make sense. So we removed this.

line 122 re-word Therefore, we deduced the absolute ...   The language should be softened - e.g. we suggest that the absolute stereochemistry of ....   based on assume biosynthetic relatedness. 

Response: Thank you for your suggestion. We re-wrote this sentence as below. “Therefore, we suggest the absolute stereostructures of 3 and 4 as shown in Figure 1 based on assume biosynthetic relatedness.”

line 131  replace "moderate" with "modest"

Response: As be pointed out, we replaced "moderate" with "modest".

line 138  replace "guessed" witih "speculate"

Response: As be pointed out, we replaced "guessed" witih "speculate".

line 160  last word "The" should not be capitalized.

Response: Thank you for your pointing out, we corrected "The" to "the".

line 162  remove one of the "collected"

Response: Thank you for your pointing out, we removed one of the "collected".

line 163  replace "its snip" with "a cutting"

Response: As be pointed out, we replaced "its snip" with "a cutting".

lines 221 & 222 what is meant by "(death)"  Suggest these be deleted.

Response: As be pointed out, we deleted it.

line 224  replace "four" with "three" as only 3 new compounds are reported - as stated in the abstract

Response: Thank you for your pointing out, we replaced "four" with "three".

line 227 replace "were identical" with "assumed to be"

As be pointed out, we replaced "were identical" with "assumed to be".

Thank you very much for the detailed review.

Reviewer 4 Report

This paper describes the isolation and characterization of a number of metabolites from a fungal species separated from a marine sponge. It seems to be the next step in a series of papers looking at other metabolites from the same source. The first compound reported has already been published without the absolute stereochemistry assigned which is corrected in this paper using CD. The three other compounds have the same basic scaffold but differing functionality present. These are all characterized by 2D NMR which forms part of the supplementary information that is unavailable for review.  The absolute stereochemistry of only compound 2 however was assigned, using a modified Mosher method. It seems the low yield of compounds 3 and 4 meant that this same technique could not be used for them. Only the relative configurations were assigned using noesy. All compounds underwent screening against a panel of cancer cell lines with modest to poor activity

The biggest concern is the ‘deduction’ of the absolute stereostructures of 3 and 4. Having the absolute stereostructure, or at least some meaningful mathematical calculations would make this a much stronger paper, why leave this to the future?

Use of the word planar to describe the 6-5-6-6 ring system. It infers that it is flat?

Line 48 the sentence ‘The previously reported the relative stereostructure of 1…’ Does not make sense- ‘The relative stereostructure of 1 has been previously reported  using DEPT…’ or something similar

Line62-65: this description is overly wordy- The main difference it the loss of a single hydroxyl group at position 1…

Line 138: use of the word guess?  

Author Response

Reviewer #4

This paper describes the isolation and characterization of a number of metabolites from a fungal species separated from a marine sponge. It seems to be the next step in a series of papers looking at other metabolites from the same source. The first compound reported has already been published without the absolute stereochemistry assigned which is corrected in this paper using CD. The three other compounds have the same basic scaffold but differing functionality present. These are all characterized by 2D NMR which forms part of the supplementary information that is unavailable for review.  The absolute stereochemistry of only compound 2 however was assigned, using a modified Mosher method. It seems the low yield of compounds 3 and 4 meant that this same technique could not be used for them. Only the relative configurations were assigned using noesy. All compounds underwent screening against a panel of cancer cell lines with modest to poor activity

The biggest concern is the ‘deduction’ of the absolute stereostructures of 3 and 4. Having the absolute stereostructure, or at least some meaningful mathematical calculations would make this a much stronger paper, why leave this to the future?

Response: Our laboratory has no means of synthesis or mathematical calculations, and we are currently looking for collaborators who can undertake these. We have softened the description of stereostructure estimation of 3 and 4. (Line 127)

Use of the word planar to describe the 6-5-6-6 ring system. It infers that it is flat?

Response: We have used “planar structure” as the description of the structure shown in Figure 3 deduced by only 1H-1H COSY and HMBC correlations. (ie, the structure that is not depicted in bold or hashed line)

Line 48 the sentence ‘The previously reported the relative stereostructure of 1…’ Does not make sense- ‘The relative stereostructure of 1 has been previously reported  using DEPT…’ or something similar

Response: As your pointed out, we re-wrote the line 48 sentence as shown in red.

Line62-65: this description is overly wordy- The main difference it the loss of a single hydroxyl group at position 1…

Response: As your pointed out, this description may be overly wordy. However, 2 is the first new compound in this paper, therefore; we described the analysis of NMR in detail, not a comparison with compound 1.

Line 138: use of the word guess?  

Response:

Response: We replaced "guessed" witih "speculate".

This manuscript is a resubmission of an earlier submission. The following is a list of the peer review reports and author responses from that submission.

Round 1

Reviewer 1 Report

1. In Results and Discussion, paragraph-1, authors mentioned about isolation of Altercrasins A-E. These structures are not even similar to trichodremanins. Please write relevant information about the isolation of trichodremanins.

2. Lines 54-56, authors mentioned about absolute configurations for C-1 & C-2 as R, R based on the CD spectrum of dibenzoate 1a. No experimental calculations or references were reported about this experiment how they performed. Please write this experiment details in supporting information.

3. Line 77, ‘replace Table S2 and Figure 4’ with Table S1. Table S2 and Figure 4 is for analysis of compound 2.

4. Line 127, figure 6, mark the rings with A, B and C.

5. The resolution of all 1D, 2D-NMR spectras which have been presented in supporting information are very bad. It would be great and easily understandable for readers, if authors put nice and well resoluted spectras.

6. I recommend to publish this work in Marine drugs.

Author Response

Reviewer #1

 1. In Results and Discussion, paragraph-1, authors mentioned about isolation of Altercrasins A-E. These structures are not even similar to trichodremanins. Please write relevant information about the isolation of trichodremanins.

Response: Thank you for your pointing out. We have posted a description of another study. We replaced that for this study.

2. Lines 54-56, authors mentioned about absolute configurations for C-1 & C-2 as R, R based on the CD spectrum of dibenzoate 1a. No experimental calculations or references were reported about this experiment how they performed. Please write this experiment details in supporting information.

Response: This method for exciton coupling and twisted pai-electron systems was established by N. Harada and co-worker as nonempirical determination of absolute stereochemistry by CD spectroscopy. We applied the methods, therefore, we added reference as No. 16 to the end of text.

3. Line 77, ‘replace Table S2 and Figure 4’ with Table S1. Table S2 and Figure 4 is for analysis of compound 2.

Response: This paragraph describes the stereochemistry of 2. So we corrected “For the stereochemistry of 1” on line 76 to “For the stereochemistry of 2”.

4. Line 127, figure 6, mark the rings with A, B and C.

Response: As be pointed out, we inserted the ring marks in figure 6

5. The resolution of all 1D, 2D-NMR spectra which have been presented in supporting information are very bad. It would be great and easily understandable for readers, if authors put nice and well resoluted spectra.

Response: Because we scanned the data on the paper media, we cannot raise the resolution any further on the update. We compensated for that by placing an enlarged view.

6. I recommend to publish this work in Marine drugs.

Response: Thank you very much.

Reviewer 2 Report

The manuscript 523153, entitled "New diterpenes with a fused 6-5-6-6 ring system isolated from the metabolites of the marine-sponge derived fungus Trichoderma harzianum”, submitted for publication in Marine Drugs, describes the isolation of new diterpenes, namely trichodermanins F–H, with a rare fused 6-5-6-6 ring system. The spectroscopic and analytical characterization by 1D- and 2D-NMR as well as by modified Mosher’s method, circular dichroism analysis and HR-FAB/MS has been also reported together with the preliminary antitumor activity in three different leukemia cell lines.

The manuscript is generally well written and the analytical/spectroscopic characterization of the compounds reported both in the experimental section and in the supplemental information is detailed enough and my interpretations of the data presented are in agreement with the structural assignments made by the Authors. Although on the chemical point of view the manuscript is interesting, I think that the major weaknesses is related to the limited novelty and scientific soundness of the data presented compared to the previous work published by the Authors (reference 12: Marine Drugs, 2017, 15, 169). In particular, in the present manuscript the Authors have determined the absolute configuration of Trichodermanins E (compound 1), which was already tentatively assigned in the previous work, by CD analysis of the corresponding dibenzoate derivative, and of Trichodermanins F (compound 2) by Mosher's method. However the absolute configuration of compounds 3 and 4 were not elucidated and only tentatively assigned/speculated by analogy with the other derivatives. Furthermore, only compounds 1 and 2 exhibited moderate cytotoxic activity in high micromolar range against leukemia cell lines (for compound 1 the cytotoxicity against the same cancer cell lines was already reported two years ago in the previous work), while compounds 3 and 4 resulted inactive. No further insights have been provided for the plausible mechanism of action neither for a rational SAR analysis of this class of compounds.

Other comments:

1. Biological activity (Table 2): It is not well clear if the data refer to cell viability or antiproliferative effect. A negative control for evaluating cytotoxicity against non tumoral cell lines should be used. Incubation time is missing. Cytotoxicity/cell viability of the vehicle (DMSO) should be clearly stated. Standard deviations should be reported. See: Leukemia Res. 1992, 16, 1165-1173; Book chapter; In vitro cytotoxicity and cell viability assays: principles, advantages, and disadvantages; DOI: 10.5772/intechopen.71923.

2. Analysis of Mosher's derivatives (Figure 5): Have the Authors performed a conformational study of Mosher's ester by quantum computational methods that can be correlated with the spectroscopic experimental findings? For example see: Eur. J. Org. Chem. 2015, 7, 1464-1471.

3. P3L64: "i.e. this compound hashad one secondary methyl.....". Please check the grammar of the sentence.

4. P4L76: "For the stereochemistry of 1". This should surely be "For the stereochemistry of 2".

5. P5L95: "The NMR spectral features (Table 1 and S3) resembledresemble". Please check the grammar of the sentence.

6. P5L126: Reference 17 does not exist in the reference section.

7. P7L179-199: It seems that something is wrong with the calculation of mol for the formation of dibenzoate 1 and mosher's esters (mol should be micromol).

Author Response

Reviewer #2

The manuscript 523153, entitled "New diterpenes with a fused 6-5-6-6 ring system isolated from the metabolites of the marine-sponge derived fungus Trichoderma harzianum”, submitted for publication in Marine Drugs, describes the isolation of new diterpenes, namely trichodermanins F–H, with a rare fused 6-5-6-6 ring system. The spectroscopic and analytical characterization by 1D- and 2D-NMR as well as by modified Mosher’s method, circular dichroism analysis and HR-FAB/MS has been also reported together with the preliminary antitumor activity in three different leukemia cell lines.

The manuscript is generally well written and the analytical/spectroscopic characterization of the compounds reported both in the experimental section and in the supplemental information is detailed enough and my interpretations of the data presented are in agreement with the structural assignments made by the Authors. Although on the chemical point of view the manuscript is interesting, I think that the major weaknesses is related to the limited novelty and scientific soundness of the data presented compared to the previous work published by the Authors (reference 12: Marine Drugs, 2017, 15, 169). In particular, in the present manuscript the Authors have determined the absolute configuration of Trichodermanins E (compound 1), which was already tentatively assigned in the previous work, by CD analysis of the corresponding dibenzoate derivative, and of Trichodermanins F (compound 2) by Mosher's method. However the absolute configuration of compounds 3 and 4 were not elucidated and only tentatively assigned/speculated by analogy with the other derivatives. Furthermore, only compounds 1 and 2 exhibited moderate cytotoxic activity in high micromolar range against leukemia cell lines (for compound 1 the cytotoxicity against the same cancer cell lines was already reported two years ago in the previous work), while compounds 3 and 4 resulted inactive. No further insights have been provided for the plausible mechanism of action neither for a rational SAR analysis of this class of compounds.

 Response: Thank you for your pointing out, and suggestion. As you pointed out, of the newly obtained new substances, only trichodermanins F (compound 2) is the one that positively determines the absolute configuration. Since we got reasonable results in the determination of the absolute structure of compound 1 that was unsolved, we listed additional studies of compound 1. For those of compounds 3 and 4, Their yields were too small to allow further experiments to determine absolute configuration. We are working on cultivating this strain to supplement them for further experiments. The cytotoxicity assay of compound 1 was again performed to confirm reproducibility and to compare under the same condition with those of compounds 2-4. In this study, we showed the preliminary antitumor activity. In the future, once we get a sufficient amount
of a derivative with potent activity, we will continue to elucidate the plausible mechanism of action including a rational SAR analysis.

Other comments:

Biological activity (Table 2): It is not well clear if the data refer to cell viability or antiproliferative effect. A negative control for evaluating cytotoxicity against non tumoral cell lines should be used. Incubation time is missing. Cytotoxicity/cell viability of the vehicle (DMSO) should be clearly stated. Standard deviations should be reported. See: Leukemia Res. 1992, 16, 1165-1173; Book chapter; In vitro cytotoxicity and cell viability assays: principles, advantages, and disadvantages; DOI: 10.5772/intechopen.71923.

Response: Thank you for your pointing out, and suggestion. Although comparison with the initial concentration of cells has not been made, we have so far reported as the cell growth inhibitory activity, since the growth ratio is calculated from comparison with the control at culture time 72 h as shown in 3.6 of Materials and Methods section. Since our experiment is a preliminary activity test on the premise that activities exhibit against both cancer and normal cell lines, we have not performed evaluation with normal cells. From now on we will study evaluating cytotoxicity against non tumoral cell lines together with cell viability. 50% cell growth inhibition rate of the vehicle (DMSO) was added to Table 2. Standard deviation cannot be shown because the original measurement data has not been saved. Please note.

Analysis of Mosher's derivatives (Figure 5): Have the Authors performed a conformational study of Mosher's ester by quantum computational methods that can be correlated with the spectroscopic experimental findings? For example see: Eur. J. Org. Chem. 2015, 7, 1464-1471.

Response: Thank you for your kind example. In our study, there is no free rotating aromatic ring around the Mosher's ester, so we believe that reasonable result has been obtained without the application of computational experiments. In the future, we will try to make computational experiments as needed to reward your suggestion.

P3L64: "i.e. this compound hashad one secondary methyl.....". Please check the grammar of the sentence.

Response: Thank you for your pointing out. The correction by the native checker remained. We corrected to “has".

P4L76: "For the stereochemistry of 1". This should surely be "For the stereochemistry of 2".

Response: Thank you for your pointing out. We corrected to “For the stereochemistry of 2".

P5L95: "The NMR spectral features (Table 1 and S3) resembledresemble". Please check the grammar of the sentence.

Response: Thank you for your pointing out. The correction by the native checker remained. We corrected to “The NMR spectral features resemble"

P5L126: Reference 17 does not exist in the reference section.

Response: Thank you for your pointing out. We added newly a reference as No.16, and as a result reference number reached 17.

P7L179-199: It seems that something is wrong with the calculation of mol for the formation of dibenzoate 1 and mosher's esters (mol should be micromol).

Response: Thank you for your pointing out. We correct to micromol.

Round 2

Reviewer 2 Report

The revised version of the manuscript did not bring any improvement. The Authors did not address any of my comments, suggestions and criticisms: standard deviations for cytotoxicity are still missing (how is possible that were not saved? Just say we didn't measure standard deviations!), a plausible hypothesis for mechanism of action is still pending, no SAR studies, no comparison with cytotoxicity in non tumoral cells was performed, absolute configurations of compounds 3 and 4 are still not elucidated. You can just compare experimental ECD spectra (you need 10 micrograms of material!) with in silico simulated electronic circular dichroism spectra obtained by time-dependent density functional theory after quantum mechanics geometrical optimization (e.g. see Mar. Drugs 2018, 16, 464 and ACS Med. Chem. Lett. 2019, 104677).

I feel reluctant to recommend this manuscript for publication cause it doesn't reach the standard quality for Marine Drugs.

Author Response

We are ashamed that we did not bring enough improvement for the points mentioned above. We could not get sample replenishment, but we addressed as much as possible.

We added standard deviations for cytotoxicity to Table 2 (page 6), and the study of the structure-activity relationship considered from the information obtained at the present time. The comparison of cytotoxicity with normal cell, and the elucidation of the detail mechanism of activity will continue to be examined after supply of these fungal metabolites.

In the absolute configuration of 3 and 4, as your suggestion, the calculate ECD method is most useful tool. But we do not have a mean to calculate ECD. We need to look for collaborators newly. We would like you to admit by adding only reference as future study subject. (page 5, line 124)